# Recruitment and Baseline Characteristics of Participants in the AgeWell.de Study—A Pragmatic Cluster-Randomized Controlled Lifestyle Trial against Cognitive Decline

**DOI:** 10.3390/ijerph18020408

**Published:** 2021-01-07

**Authors:** Susanne Röhr, Andrea Zülke, Melanie Luppa, Christian Brettschneider, Marina Weißenborn, Flora Kühne, Isabel Zöllinger, Franziska-Antonia Zora Samos, Alexander Bauer, Juliane Döhring, Kerstin Krebs-Hein, Anke Oey, David Czock, Thomas Frese, Jochen Gensichen, Walter E. Haefeli, Wolfgang Hoffmann, Hanna Kaduszkiewicz, Hans-Helmut König, Jochen René Thyrian, Birgitt Wiese, Steffi G. Riedel-Heller

**Affiliations:** 1Institute of Social Medicine, Occupational Health and Public Health (ISAP), Medical Faculty, University of Leipzig, 04103 Leipzig, Germany; andrea.zuelke@medizin.uni-leipzig.de (A.Z.); melanie.luppa@medizin.uni-leipzig.de (M.L.); Steffi.Riedel-Heller@medizin.uni-leipzig.de (S.G.R.-H.); 2Global Brain Health Institute (GBHI), Trinity College Dublin, D02 PN40 Dublin, Ireland; 3Department of Health Economics and Health Services Research, University Medical Center Hamburg-Eppendorf, 20246 Hamburg, Germany; c.brettschneider@uke.de (C.B.); h.koenig@uke.de (H.-H.K.); 4Department of Clinical Pharmacology and Pharmacoepidemiology, University Hospital Heidelberg, 69120 Heidelberg, Germany; marina.weissenborn@med.uni-heidelberg.de (M.W.); david.czock@med.uni-heidelberg.de (D.C.); walter.emil.haefeli@med.uni-heidelberg.de (W.E.H.); 5Institute of General Practice/Family Medicine, University Hospital of LMU Munich, 80336 Munich, Germany; flora.kuehne@med.uni-muenchen.de (F.K.); isabel.zoellinger@med.uni-muenchen.de (I.Z.); Jochen.Gensichen@med.uni-muenchen.de (J.G.); 6Institute of General Practice and Family Medicine, Martin-Luther-University Halle-Wittenberg, 06112 Halle (Saale), Germany; franziska.samos@medizin.uni-halle.de (F.-A.Z.S.); alexander.bauer@medizin.uni-halle.de (A.B.); thomas.frese@uk-halle.de (T.F.); 7Institute of General Practice, University of Kiel, 24105 Kiel, Germany; j.doehring@allgemeinmedizin.uni-kiel.de (J.D.); kkh@allgemeinmedizin.uni-kiel.de (K.K.-H.); kaduszkiewicz@allgemeinmedizin.uni-kiel.de (H.K.); 8Institute for General Practice, Work Group Medical Statistics and IT-Infrastructure, Hannover Medical School, 30625 Hannover, Germany; oey.anke@mh-hannover.de (A.O.); wiese.birgitt@mh-hannover.de (B.W.); 9Institute for Community Medicine, University Medicine Greifswald, 17489 Greifswald, Germany; wolfgang.hoffmann@uni-greifswald.de (W.H.); rene.thyrian@dzne.de (J.R.T.); 10German Center for Neurodegenerative Diseases (DZNE), Site Rostock/Greifswald, 17489 Greifswald, Germany

**Keywords:** AgeWell.de, dementia, cognitive function, prevention, intervention, lifestyle, trial, RCT, cluster-randomized controlled trial, primary care, general practitioner

## Abstract

Targeting dementia prevention, first trials addressing multiple modifiable risk factors showed promising results in at-risk populations. In Germany, AgeWell.de is the first large-scale initiative investigating the effectiveness of a multi-component lifestyle intervention against cognitive decline. We aimed to investigate the recruitment process and baseline characteristics of the AgeWell.de participants to gain an understanding of the at-risk population and who engages in the intervention. General practitioners across five study sites recruited participants (aged 60–77 years, Cardiovascular Risk Factors, Aging, and Incidence of Dementia/CAIDE dementia risk score ≥ 9). Structured face-to-face interviews were conducted with eligible participants, including neuropsychological assessments. We analyzed group differences between (1) eligible vs. non-eligible participants, (2) participants vs. non-participants, and (3) between intervention groups. Of 1176 eligible participants, 146 (12.5%) dropped out before baseline; the study population was thus 1030 individuals. Non-participants did not differ from participants in key sociodemographic factors and dementia risk. Study participants were M = 69.0 (SD = 4.9) years old, and 52.1% were women. The average Montreal Cognitive Assessment/MoCA score was 24.5 (SD = 3.1), indicating a rather mildly cognitively impaired study population; however, 39.4% scored ≥ 26, thus being cognitively unimpaired. The bandwidth of cognitive states bears the interesting potential for differential trial outcome analyses. However, trial conduction is impacted by the COVID-19 pandemic, requiring adjustments to the study protocol with yet unclear methodological consequences.

## 1. Introduction

A growing body of evidence highlights the enormous potential for dementia prevention if modifiable risk factors were addressed. In fact, it is estimated that 40% of all cases of dementia in high-income countries (HIC) and 50% in low- and middle-income countries (LMIC) could be prevented or at least delayed if the following twelve risk factors were eliminated: low education in early life, hearing loss, traumatic brain injury, hypertension, obesity, alcohol consumption above 21 units a week in midlife, diabetes mellitus, depression, physical inactivity, smoking, social isolation, and exposure to air pollution in later life [1]. There is potential for the identification of further health, lifestyle, and environmental factors for cognitive decline and dementia in the future, which may increase the estimate of potentially preventable dementia cases.

The dementia prevention capacity provides hope in the face of population aging all around the world, which is associated with a significant increase in the number of people living with dementia, specifically in LMIC. While currently, over 50 million people are living with dementia; this number is expected to increase to more than 152 million by 2050 [2]. Already, dementia is the main cause of disability among older individuals, and it comes with a heavy burden for everyone involved: patients, relatives, and caregivers, and societies as a whole, causing costs equivalent to 1% of the global world product [3,4]. Since most types of dementia cannot be cured (yet), prevention is key in counteracting the dementia epidemic [4]. Studies investigating secular trends in dementia incidence provide further evidence of dementia prevention potential: They observed declining numbers in new dementia cases in more recent birth cohorts, which is most likely due to increased levels of education and improved management of cardiovascular risk factors [5,6,7].

Going beyond the observational evidence, an increasing number of randomized controlled trials (RCT) investigates the effectiveness of complex lifestyle interventions, simultaneously addressing multiple modifiable risk factors of cognitive decline and dementia. The pioneer study, the Finnish Geriatric Intervention Study to Prevent Cognitive Impairment and Disability (FINGER), was able to show a significant benefit of a multi-domain lifestyle intervention on cognitive function over two years [8]. Other lifestyle-based trials from Europe, namely the French Multi-domain Alzheimer Preventive Trial (MAPT) and the Dutch Prevention of Dementia by Intensive Vascular Care (Pre-DIVA) trial, have been less conclusive but suggested benefits for cognitive function in specific subgroups of participants with increased risk for dementia [9,10,11]. These promising but still inconsistent results have led to an initiative of joint forces of similar initiatives around the globe: World-Wide FINGERS (WW-FINGERS) currently brings together lifestyle trials from over 25 countries, including protocols from symptomatic states to early symptomatic stages of dementia and covering diverse geographical, cultural, and economic settings [12].

To this end, the AgeWell.de study is the first trial in Germany, aiming to evaluate the effectiveness and cost-effectiveness of a multi-component lifestyle intervention against cognitive decline in older primary care patients at increased dementia risk. Drawing and extending on the FINGERS model, our multi-component intervention includes nutritional counseling, physical activity enhancement, cognitive training, monitoring of vascular risk factors, optimization of medication, social activity enhancement, and, if necessary, interventions targeting depression, bereavement, and grief. Here, we aimed to provide a report on (1) the recruitment process and (2) key baseline characteristics regarding sociodemographic, health, and lifestyle factors, as well as the cognitive function of the study sample and with respect to the intervention groups. Understanding the at-risk population and understanding who is participating in lifestyle trials against cognitive decline is important for the interpretation of the trial results, and it may inform future studies aiming to develop lifestyle interventions.

## 2. Materials and Methods

### 2.1. Study Design

AgeWell.de is designed as a multi-centric, cluster-RCT, with two arms, which was conducted at five study sites in five German urban areas (Leipzig, Kiel, Greifswald, Munich, and Halle). The aim is to deliver a multi-component lifestyle intervention, targeting modifiable health and lifestyle factors of dementia, to primary care patients at increased dementia risk over an intervention period of two years. The effectiveness of the intervention is evaluated against general health advice/treatment as usual (control condition), targeting cognitive function (primary outcome) and measures of mental and physical health (secondary outcomes). Additionally, the Department of Health Economics and Health Service Research at the University Medical Center Hamburg-Eppendorf evaluates the cost-effectiveness of the trial. Furthermore, the Department of Clinical Pharmacology and Pharmacoepidemiology at the University Hospital Heidelberg manages the intervention component “optimization of medication”, which includes the evaluation of electronic data on participants’ medication regimes in order to provide feedback to the general practitioner (GP). The Institute for General Practice at the Hannover Medical School is the responsible site for decentralized data management, which oversees the internet-based remote data capture system.

The AgeWell.de trial is registered in the German Clinical Trials Register (DRKS; trial identifier: DRKS00013555). The study is currently ongoing. A study protocol detailing the study design is available [13].

### 2.2. Ethics

The study was approved by the responsible ethics boards of the coordinating center (Ethics Committee of the Medical Faculty of the University of Leipzig; ethical vote number: 369/17-ek) and of all participating study sites and expertise centers. Participants provided written informed consent to participate in the GPP (general practitioner practice).

### 2.3. Recruitment Procedure

Community-dwelling GP (general practitioner) patients were recruited from June 2018 to October 2019 by GPP in the areas of the participating study sites. To be included, individuals had to be between 60 and 77 years old and at increased dementia risk, quantified by the Cardiovascular Risk Factors, Aging, and Incidence of Dementia (CAIDE) score [14]. A cut-off of ≥9 points was defined for the inclusion of participants since this score predicted dementia risk with a sensitivity of 0.77 and a specificity of 0.63 in previous studies [14]. CAIDE is based on information that is easy to assess (age, education, gender, blood pressure, body mass index, total cholesterol, and physical activity), facilitating case finding of eligible participants in the GPP. Exclusion criteria were dementia diagnosed or suspected by the GP, medical conditions potentially affecting safe engagement in the intervention (malignant disease/fatal illness, severe clinical depression, symptomatic cardiovascular disease, revascularization within the previous year) as judged by the GP, severe loss of vision, hearing, or communicative ability/insufficient ability to speak and read German, severe mobility impairment, and coincident participation in another intervention trial.

### 2.4. Intervention

Guided by the FINGER model, the multi-component intervention comprises nutritional counseling, physical activity enhancement, cognitive training, and monitoring of vascular risk factors during the 2-year intervention period. Moreover, our study addresses additional established risk factors: social inactivity [14,15] and potentially inappropriate medication (e.g., anticholinergic drugs) or medication overuse and underuse [16]. Therefore, feedback and specific recommendations on participants’ medication are provided to their attending GPs, if indicated. Finally, we address further known risk factors for dementia by providing specific interventions in the case of bereavement, grief, and depressive symptoms, if applicable. Intervention A (advanced) was administered in a tailored, individualized manner during face-to-face sessions by trained study personnel during a visit to the participants’ homes. Participants of intervention group B (basic) received GP treatment as usual (GPTAU) and general health advice (GHA), covering the components of intervention A. It was hypothesized that the multi-component intervention program would be superior to GPTAU and GHA regarding trial outcomes. As detailed information on the intervention is less relevant for this report, we would like to point to the study protocol for a comprehensive description of the intervention [13].

### 2.5. Sample Size

The sample size was calculated based on results of previous studies [17], assuming a mean decrease of *z* = −0.21 points (SD = 0.5) in a composite cognitive test score in intervention group B over the two-year intervention period. In order to detect a 50% difference in change in the composite z-score between the two intervention groups, a sample size of 475 participants was required in each group, with a level of significance of 5% and 90% power. In light of the findings from the FINGER, the dropout rate is expected not to exceed 10% [8]. Therefore, an optimal sample size would comprise 1152 participants (576 per group).

### 2.6. Randomization and Blinding

Block-randomization of the GPP was applied to ensure a balance in sample size for the two groups, intervention A and B (ratio 1:1). Randomization was handled by the data management center at the Institute for General Practice at the Hannover Medical School, using a randomization list that was concealed from the study sites. Participating GPPs were blinded to their group allocation (intervention A or B).

### 2.7. Baseline Assessment

Structured face-to-face interviews were conducted with all participants at baseline, assessing the reference for the primary and secondary trial outcomes as well as other relevant information (e.g., sociodemographic information, lifestyle, and health-related factors). Moreover, GPP provided further information on participants’ health characteristics (medical diagnoses, lab values, medication) using standardized questionnaires. Assessments relevant for this report included the following:

Cognitive performance was assessed using a neuropsychological test battery, covering six cognitive domains for diagnosing mild and/or major neurocognitive disorder according to DSM-5 (attention, executive function, learning/memory, language, perceptual-motor abilities, and social cognition). The respective domains were assessed by (1) the Trail Making Test A for attention [18], (2) Trail Making Test A and B for executive function [18], (3) the Consortium to Establish a Registry for Alzheimer’s Disease (CERAD) subtest “Word List Memory”-subtest for learning/memory [19,20,21], (4) CERAD subtest “Verbal Fluency Test (Animals)” for language [19,20,22,23], (5) CERAD subtest “Constructional Praxis” assessing perceptual-motor skills [19,20,24], and (6) the Reading the Mind in the Eyes-Test (revised version) for social cognition [25,26]. Additionally, overall cognitive performance was assessed using the Montreal Cognitive Assessment (MoCA [27]; permission to use the MoCA in the trial was obtained from the copyright owners).

Further baseline assessments included (instrumental) activities of daily living (ADL/IADL, assessed by the Barthel-Index [28] and the Amsterdam IADL-scale [29], respectively), depressive symptoms (assessed with the Geriatric Depression Scale [30]), social isolation (assessed with the short form of Lubben Social Network Scale/LSNS-6 [31]), anthropometric measures (height, weight, body mass index), and blood pressure. Information on nutrition was assessed using a validated food frequency questionnaire [32], assessing volume and frequency of consumption of 53 specific foods and beverages, including information on consumption of fruit, vegetables, and alcohol. Moreover, we assessed physical activity (assessed using self-constructed items), bereavement, and grief (assessed using a standardized instrument for the assessment of bereavement [33]). GPP provided further information on participants’ medical diagnoses, using standardized questionnaires.

### 2.8. Statistical Analyses

Differences between eligible and non-eligible participants, participants and non-participants, and between trial participants in intervention A vs. B were analyzed using Chi-square tests or t-tests, as appropriate. Figures were presented as mean and SD values or proportions. A *p*-value of 0.05 was determined to indicate the significance of differences. Analyses were conducted using Stata 16 (SE; StataCorp., College Station, TX, USA).

## 3. Results

### 3.1. Recruitment and Sample Selection

The recruitment process and sample selection are displayed in Figure 1. Of 840 GPP that were initially contacted, 201 (23.9%) expressed their interest to participate; 150 (17.9%) practices provided written informed consent to participate. Of those, 27 (3.2%) practices never recruited any participant (e.g., due to unforeseen time constraints), leaving 123 (14.6%) practices. These were randomized to either intervention A (*n* = 64) or intervention B (*n* = 59). Following the consent to participate, respective GPP were trained in all recruitment procedures by AgeWell.de-study personnel during an appointment at the GPP. Training included the attending GP and, if desired, further practice personnel. Case findings were then conducted by the GP or other practice personnel in association with the AgeWell.de-study personnel or independently if desired. For details regarding the recruitment process, please see [13].

A total of 1176 patients were recruited and consented to participate, i.e., provided written informed consent at the GPP, and underwent subsequent screening for eligibility. Of those, 44 (3.7%) were not eligible, mostly due to non-fulfillment of the cutoff of the CAIDE score, and 102 (8.7%) dropped out before baseline assessment due to the occurrence of severe health problems, relocation, or withdrawal of consent. This resulted in baseline interviews of 1030 patients (intervention A vs. intervention B: *n* = 487 vs. 543), constituting the AgeWell.de study population.

#### 3.1.1. Eligible vs. Non-Eligible Participants

Table 1 provides information on key characteristics of eligible vs. non-eligible participants of the AgeWell.de trial. Non-eligible participants did not differ from participants regarding age and sex, but they had significantly more years of education. As per inclusion criteria, dementia risk (CAIDE-score) was higher in eligible participants than in non-eligible participants.

#### 3.1.2. Participants vs. Non-Participants

Table 2 provides information on key characteristics of participants and non-participants of the AgeWell.de trial. Non-participants did not differ from participants regarding age, sex, and education. Moreover, dementia risk (CAIDE-score) did not differ between groups. Regarding the single domains of the CAIDE score, non-participants had significantly less often systolic blood pressure levels >140 mmHg than participants; however, they did not differ with regards to other CAIDE measures.

#### 3.1.3. Baseline Characteristics of the Study Sample

Baseline characteristics of the 1030 study participants of the AgeWell.de trial are provided in Table 3. Their mean age was 69.0 (SD = 4.9) years; 52.1% were women. The average duration of the baseline interview was 137.1 min (SD = 35.8) with slight differences between intervention groups A and B (M = 139.9, SD = 33.0; M = 134.5, SD = 37.9, respectively; *p* = 0.016). About half of the sample had a medium level of education (53.0%), and about a quarter each had a low (24.4%) or high level (22.6%), as assessed by the CASMIN-classification (Comparative Analysis of Social Mobility in Industrial Nations [34]), which is based on information on general and vocational education. Almost two thirds (64.6%) were married or in a partnership. Participants in intervention group A did not differ from those in intervention group B with regards to sociodemographic factors. The total MoCA score was 24.5 (SD = 3.1), with no difference between groups. Intervention group A only exhibited slightly better results on the CERAD Word List Learning (M = 19.2, SD = 4.1 vs. M = 18.5, SD = 4.9; *p* = 0.024) and CERAD Word List Recall tests (M = 5.9, SD = 2.2 vs. M = 5.6, SD = 2.3; *p* = 0.042) compared to intervention group B. There were no group differences with regard to health and lifestyle factors, except for higher systolic blood pressure/SBP (prevalence: 59.0% vs. 49.3%; *p* = 0.002; mmHg: M = 147.2, SD = 18.4 vs. M = 144.1, SD = 19.9; *p* = 0.012) in intervention group A compared to B.

## 4. Discussion

AgeWell.de is the first trial in Germany, investigating the effectiveness and cost-effectiveness of a multi-component lifestyle intervention on cognitive function in primary care patients at increased risk for dementia. It is designed as a pragmatic trial to allow for rapid implementation into routine care in Germany if trial results indicate success. Moreover, as the AgeWell.de trial expands on the intervention components of the FINGER study, we expect to contribute new knowledge on additional factors to the growing international network of lifestyle trials to work towards effective dementia prevention.

Inspection of the recruitment process and baseline characteristics of the AgeWell.de participants provided useful information on the study progress and the population under investigation. While the overall recruitment and allocation strategy were largely successful, there were deviations between the targeted and actual number of GPP as well as study participants. Regarding GPP, it was anticipated that 96 practices would be sufficient to recruit the calculated sample of 1152 individuals, which would require each GPP to recruit 12 participants. However, it would soon become clear that not every GPP would be able to reach the target, e.g., due to time constraints or too few eligible patients. This led to a need for recruiting more GPP than anticipated and an unequal number of patients per GPP, resulting in a participant group allocation ratio that was not 1:1. Moreover, even though GPPs were able to identify 1176 eligible patients who provided written informed consent, 146 of them could not be included in the trial (44 did not fulfill the inclusion criteria, specifically the CAIDE score, and 102 of them changed their mind about participation), leading to a study participation pool of 1030. This was not expected to this extent. But initial power calculations were based on conservative estimates. With the actual numbers of participants at baseline being 487 and 543 in the intervention groups A and B, respectively, the deviation from the targeted number should not interfere with the primary and secondary study objectives. However, an even stronger emphasis is, therefore, put on the continuous motivation of participants and on monitoring intervention adherence. All study personnel have been trained in motivational interview techniques and are in a continuous exchange over potential problems with participants to keep dropout low and adherence high.

Putting the baseline characteristics of the AgeWell.de study population in context revealed several similarities with populations from previous lifestyle interventions. The mean age of our participants was highly comparable to that of the FINGER trial (69.4 years), which applied the same age range. Results in several cognitive tests were also comparable to FINGER (CERAD Wordlist Learning, mean: 18.4; CERAD Wordlist Recall, mean: 5.5 [35]) or the MAPT trial (Trail Making Test B, mean: 121.5 [9]). Mean SBP was similar to the values reported in other trials (FINGER: 140.1 mmHg [34]; MAPT: 141.0 mmHg [9]), while the proportion of participants with body mass index (BMI) ≥ 30 kg/m^2^ was higher in our study sample (FINGER: 29.9% [35]; MAPT: 26.1% [9]; Pre-DIVA: 27.7% [36]). The proportion of individuals with hypertension was higher than in the FINGER trial (65.9% [35]); however, while FINGER used self-report information for the assessment of diagnoses, the information in AgeWell.de was provided by the attending GPP. Moreover, differences were likely associated with varying CAIDE cutoff scores between studies (AgeWell.de: ≥9 points; FINGER: ≥6 points), which was chosen to be more conservative in AgeWell.de to target a population high at risk for dementia. Therefore, it is plausible that certain conditions, increasing risk for dementia, were more frequent in our sample.

The prevalence for hypertension in the German general population, as reported in the German Health Interview and Examination Study for Adults (DEGS1), was 59.8% in the age group 60–69 years and 74.2% in the 70 to 79 years olds [37]. Prevalence of obesity (defined as BMI ≥ 30 kg/m^2^) was also higher in the AgeWell.de-study sample than in the German general population (women: 34.8 in the age group 60–69; 41.6 in the age group 70–79; men: 33.1 in the age group 60–69; 31.3 in the age group 70–79; [38]). These differences were likely due to our recruitment strategy using the CAIDE Dementia Risk Score, which includes high blood pressure and BMI as risk factors. While 13.2% of study participants indicated to be current smokers, the respective value in the German general population aged 65–79 years is 11.5% [39]. Rates for the history of myocardial infarction were lower than in the general population (60–69 years: 8.2%; 70–79 years: 10.2%), while rates for coronary heart disease were slightly higher than in the general population of the same age (60–69 years: 15.1%; 70–79 years: 22.3%; [40]).

Considering cognitive performance, the overall average MoCA score of 24.5 indicated that the AgeWell.de study population was rather mildly cognitively impaired as the original cutoff score for mild cognitive impairment (MCI) was established at <26 [27]. Though this cutoff has been found to be rather conservative, comparison with normative values of the MoCA for the German general old age population would likewise suggest that the average study participant of AgeWell.de was mildly cognitively impaired at baseline, with scores being slightly below age, sex, and education-specific norms [41]. However, 405 (39.4%) participants scored in the range of 26 to 30 points, indicating no cognitive impairment, and 622 participants fell below 26 points, thus indicating probable MCI. This will allow for interesting differential trial outcome analyses with respect to the baseline cognitive status.

### 4.1. Limitations

To keep the workload for GPP low, we only collected very little information in regard to initially approached patients, which is why we limited comparison between eligible vs. non-eligible participants as well as participants vs. non-participants. However, regarding the non-participation, there was no hint towards group differences, except for slightly higher systolic blood pressure in the participants compared to non-participants.

AgeWell.de population is a selected sample that was drawn based on increased dementia risk. This limits the generalizability of the trial results in dementia risk populations based on pre-specified sociodemographic and vascular factors, which have to be considered when implementation strategies will be developed. Moreover, as the AgeWell.de population was predominantly drawn from urban areas, a certain clarity of the applicability of the intervention to populations from rural areas remains.

Dementia prevention is a rapidly evolving topic with emerging evidence for further modifiable risk factors that could be targeted in future interventions against cognitive decline and dementia, for example, hearing impairment, traumatic brain injury, and exposure to air pollution. Such factors could not have been targeted in the AgeWell.de study yet.

### 4.2. Outlook

AgeWell.de is currently ongoing, with trial results being expected for early 2022. The emergence of the COVID-19 pandemic in early 2020 has imposed unprecedented challenges in trial conduction with currently unknown impact. While the recruitment and baseline assessments were completed before the pandemic, imposed public health measures, including lockdowns, to curb the spread of SARS-CoV-2 from late March 2020 largely overlap with the intervention period, potentially interfering with the way study participants in intervention group A are able to carry out and adhere to some of the interventional components (e.g., social and physical activity). However, a postal survey measuring the impact of the pandemic on everyday life and social and mental health was carried out during the first lockdown among all AgeWell.de participants, and the perceived impact on the intervention components was assessed among intervention A participants, specifically. A repeated survey to monitor the COVID-19 impact will follow.

Moreover, a 2-year follow-up face-to-face assessments, which commenced in early fall 2020, needed to be adapted to meet strict hygiene protocols to ensure the highest possible infection protection for study participants as well as study personnel. Adaptation to a different assessment mode in order to entirely avoid face-to-face contact is not possible without violating data integrity with respect to establishing trial outcomes based on data comparisons with baseline assessments. With a resurgence of COVID-19 cases in fall 2020 and further lockdowns, interruptions in the 2-year follow-up assessments are very likely. Plans on how to meaningfully account for the COVID-19 impact in trial outcome analyses have yet to be worked out. Moreover, the willingness of study participants to participate in face-to-face assessments may be lower because of COVID-19. In this case, study participants will be offered a telephone interview that will allow for comprehensive dropout analysis.

## 5. Conclusions

The AgeWell.de study, the first lifestyle trial against cognitive decline in Germany, will add important knowledge to the growing field of dementia prevention based on risk factor modification as well as implementation opportunities in primary care settings. The recruitment process and baseline assessments have been completed, the randomization strategy was successful, and baseline data confirmed high dementia risk load among a study population with diverse cognitive states ranging from cognitive healthy to MCI, bearing potential for differential trial outcome analyses in this regard. Moreover, our findings may inform the study design of future lifestyle trials. However, the COVID-19 pandemic has introduced unprecedented practical and methodological challenges in the trial conduction, and consequences for trial outcomes have yet to be discussed. Careful monitoring of the impact of the COVID-19 pandemic on study participants’ lifestyles needs to complement the study protocol.

## Figures and Tables

**Figure 1 ijerph-18-00408-f001:**
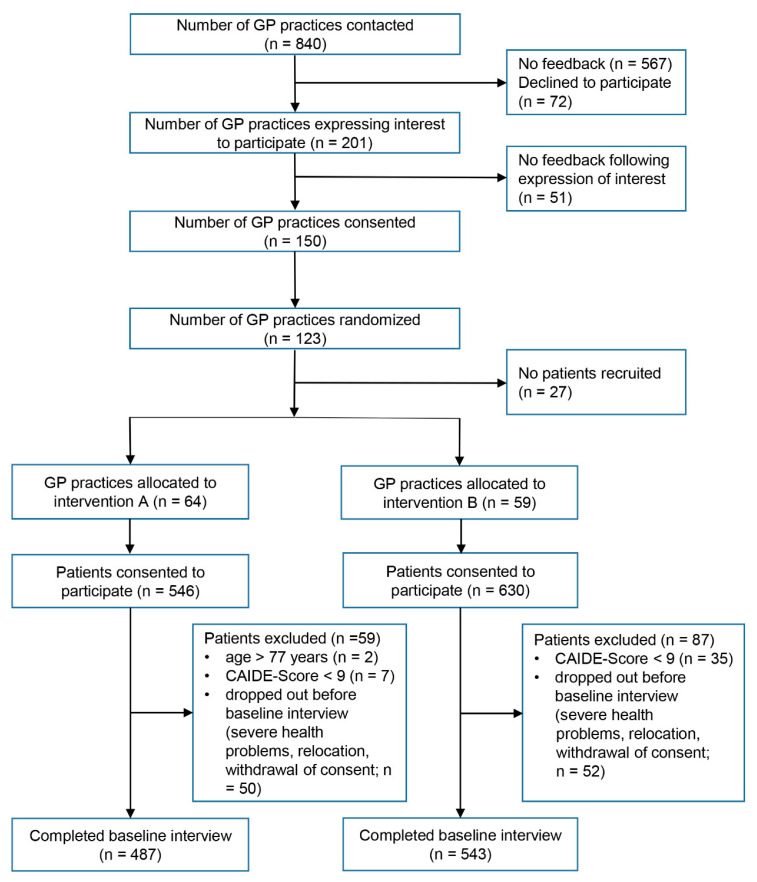
Recruitment process and sample selection of participants in the AgeWell.de trial.

**Table 1 ijerph-18-00408-t001:** Characteristics of eligible and non-eligible participants in the AgeWell.de trial.

Variables	Overall (n = 1176)	Eligible (n = 1132)	Non-Eligible (n = 44)	*p*-Value
Age in years, mean (SD)	69.0 (4.9)	69.0 (4.9)	69.3 (5.7)	0.765
Sex, female (%)	52.0	52.1	47.7	0.567
Education, years (%)				
≥10	54.4	53.5	78.6	
7–9	44.2	45.1	21.4	
<7	1.4	1.4	0	0.006
CAIDE-Dementia Risk Score (points); mean (SD)	10.1 (1.3)	10.2 (1.2)	7.4 (1.9)	<0.001
Systolic blood pressure > 140 mmHg (%)	85.0	85.6	70.5	0.006
Total cholesterol > 6.5 mmol/L (%)	53.8	55.2	18.2	<0.001
Body mass index > 30 kg/m² (%)	57.5	59.4	7.0	<0.001
Physical activity at least 2 times/week at least 30 minutes (%)	25.6	24.8	46.5	0.001
Intervention A (%)	46.4	47.4	20.5	<0.001

**Table 2 ijerph-18-00408-t002:** Characteristics of participants and non-participants of the AgeWell.de trial.

Variables	Participants (n = 1030)	Non-Participants (n = 102)	*p*-Value
Age in years, mean (SD)	69.0 (4.9)	69.7 (4.6)	0.125
Sex, female (%)	52.1	52.0	0.973
Education, years (%)			
≥10	54.5	44.1	
7–9	44.3	52.9	
<7	1.3	2.9	0.072
CAIDE-Dementia Risk Score (points); mean (SD)	10.2 (1.2)	10.3 (1.3)	0.421
Systolic blood pressure > 140 mmHg (%)	86.8	73.5	<0.001
Total cholesterol > 6.5 mmol/L (%)	55.5	52.9	0.628
Body mass index > 30 kg/m^2^ (%)	58.7	66.3	0.138
Physical activity at least 2 times/week at least 30 min (%)	25.2	21.6	0.425
Intervention A (%)	47.3	49.0	0.737

**Table 3 ijerph-18-00408-t003:** Baseline characteristics of study participants in the AgeWell.de trial with respect to group allocation (*n* = 1030).

Variables	*n*	Overall (n = 1030)	Intervention A (n = 487)	Intervention B (n = 543)	*p*-Value
Duration of interview in minutes, M (SD)	1030	137.1 (35.8)	139.9 (33.0)	134.5 (37.9)	0.016
Sociodemographic characteristics					
Age in years, mean (SD)	1030	69.0 (4.9)	69.1 (4.9)	68.8 (5.0)	0.418
Sex, female (%)	1030	52.1	52.0	52.3	0.910
Education (CASMIN), %	1030				
Low		24.4	27.3	21.7	
Intermediate		53.0	52.6	53.4	
High		22.6	20.1	24.9	0.053
Income in Euros, mean (SD)	930	1572.2 (845.8)	1543.1 (779.0)	1597.8 (900.7)	0.325
Married/cohabitating, %	1030	64.6	65.5	63.7	0.550
Cognitive function					
MoCA sum score (points), mean (SD)	1026	24.5 (3.1)	24.5 (3.0)	24.6 (3.1)	0.574
Total score ≥ 26, %		39.4	36.8	41.8	
Total score < 26, %		60.6	63.2	58.2	0.106
CERAD: Word List Learning (words), mean (SD)	1004	18.8 (4.5)	19.2 (4.1)	18.5 (4.9)	0.024
CERAD: Word List Recall (words), mean (SD)	1002	5.7 (2.3)	5.9 (2.2)	5.6 (2.3)	0.042
CERAD: Verbal Fluency (words), mean (SD)	1022	22.2 (5.6)	21.9 (5.5)	22.5 (5.7)	0.114
Trail Making Test A (seconds), mean (SD)	1012	52.3 (21.2)	52.5 (20.9)	52.2 (21.5)	0.824
Trail Making Test B (seconds), mean (SD)	1000	123.5 (59.7)	122.4 (59.7)	124.5 (59.8)	0.582
Reading the Mind in the Eyes Test (points), mean (SD)	975	20.6 (4.1)	20.5 (4.1)	20.8 (4.1)	0.257
Health and lifestyle factors					
High blood pressure (%)					
SBP > 140 mmHg	1012	53.9	59.0	49.3	0.002
DBP > 90 mmHg	1012	29.1	30.0	28.2	0.528
Systolic blood pressure, (mmHg), mean (SD)	1024	145.6 (19.2)	147.2 (18.4)	144.1 (19.9)	0.012
Diastolic blood pressure, (mmHg), mean (SD)	1024	85.5 (11.0)	86.1 (10.4)	84.9 (11.5)	0.081
BMI (kg/m^2^), mean (SD)	1014	31.0 (5.5)	31.0 (5.3)	31.0 (5.6)	0.998
BMI ≥ 30 kg/m^2^ (%)	1014	55.3	55.9	54.8	0.869
Geriatric Depression Scale (points), mean (SD)	1016	1.6 (2.0)	1.6 (1.9)	1.7 (2.1)	0.381
Lubben Social Network Scale (points), mean (SD)	1025	17.1 (5.6)	17.4 (5.6)	16.8 (5.7)	0.073
Instrumental Activities of Daily Living (A-IADL; points), mean (SD)	1024	0.05 (0.15)	0.04 (0.13)	0.06 (0.16)	0.181
Activities of Daily Living (Barthel-Index; points), mean (SD)	1028	99.5 (3.1)	99.5 (3.0)	99.5 (3.1)	0.906
Current smoker (%)	975	13.2	14.3	12.3	0.369
Alcohol drinking ≥ once a week (%)	889	60.0	59.4	60.5	0.741
Fruit consumption in grams/day, mean (SD)	1000	386.7 (412.1)	392.7 (403.7)	381.4 (419.8)	0.667
Vegetable consumption in grams/day, mean (SD)	998	140.1 (141.1)	135.1 (137.1)	144.6 (144.6)	0.288
GP diagnoses					
Depression (%)	1022	11.6	9.7	13.3	0.074
Diabetes (%)	1025	39.5	37.9	41.0	0.304
History of myocardial infarction (%)	1016	5.9	6.1	5.8	0.849
History of stroke (%)	1019	4.4	4.1	4.7	0.665
Hypercholesterolemia /Hyperlipidemia (%)	1014	72.5	72.8	72.2	0.850
Renal insufficiency/chronic kidney disease (%)	1012	15.9	13.7	18.0	0.062
Hypertension (%)	1023	87.5	89.0	86.1	0.166
Coronary heart disease (%)	1020	17.3	17.4	17.1	0.913
Obesity (BMI ≥ 30 kg/m^2^) (%)	1023	54.4	57.5	51.5	0.053

Abbreviations: A-IADL: Amsterdam Instrumental Activity of Daily Living *Questionnaire*; BMI: body mass index; CASMIN: Comparative Analysis of Social Mobility in Industrial Nations; CERAD: Consortium to Establish a Registry for Alzheimer’s Disease; DBP: diastolic blood pressure; MoCA: Montreal Cognitive Assessment; SBP: systolic blood pressure; SD: standard deviation; GP: general practitioner.

## Data Availability

The data presented in this study are available on request from the corresponding author. The data are not publicly available yet as the trial is currently ongoing.

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
