# Peer review of "Recruitment and Baseline Characteristics of Participants in the AgeWell.de Study—A Pragmatic Cluster-Randomized Controlled Lifestyle Trial against Cognitive Decline"

_ijerph, 2021, doi:10.3390/ijerph18020408_

Round 1
Reviewer 1 Report
1. The manuscript appears to be a report of an ongoing study which has encountered barriers related to recruitment, adherence, and assessment in part related to pandemic effects on scientific research. It best fits the category of a feasibility report. The title should orient the reader, i.e.: A feasibility report: Recruitment and baseline characteristics of participants.......
2. The MOCA is now a licensed test and permission is required for its use in research since September 1, 2019. While the research began in June 2018, it is an ongoing study and the authors are advised to consult <mocatest.org/permission/> for guidance.
3. The authors describe the study design and the recruitment population including planned intervention and control groups and their baseline assessments.
4. The study is an ongoing intervention designed as a two-year cluster RCT.
5. A results section should be included. This could discuss the barriers in achieving planned goals, i.e. differences among primary care practices, reasons for dropouts, pandemic effects on intervention, adherence, and assessment of subjects.
6. The discussion and conclusions could generalize on the pandemic effects on conduct of clinical research and anticipated adaptations required to achieve study goals. This information may be useful to other investigators and enhance the benefit of the manuscript.
Author Response
Please see document attached.

Reviewer 2 Report
In this paper the authors present the recruitment and baseline characteristics of participants in the AgeWell.de study. The authors should be commended for their effort to perform a lifestyle based trial to hopefully achieve a reduction dementia prevalence. Especially in a time of COVID this will be a difficult trial.
As the purpose of this manuscript is to publish the recruitment, I will not comment on the study design and rationale itself. I have several questions regarding the recruitment process and theg generalizability of the study findings.
- The methods now say that 'dementia suspected by the GP' was a reason for exclusion. However, did the GP's use any study definition for this? And was the MOCA score then used for this definition? A GP might not be experienced enough to define or diagnose dementia. Besides, was a low MOCA score an exclusion criteria?
- Several GP practices were reached and included for this study. However those were al GP's in a city. THis might influence your study population and limit the generalizability toward rural GP practices and patients. Could the authors comment on this?
- What about etnicity and migrants in this study? Will the study also include patients with diverse etnicity? Or perform for example a sub analysis?
- How did you handle possible language barrieres, as an exclusion critiera?
- Table 3 shows the GP diagnoses. The Table does not show Chronic Kidney Disease. I would strongly suggest to incorporate CKD in your study as an important cognition/ dementia-related variable.
- Could the authors provide information on medication use in Table 3? The methods state that information on medication was collected (Page 5, line 189).
- Finally, in line 64 it is mentioned that air pollution is also an important risk factor. This might be an important variable in your 'city study population'. However, it is hard to change this risk factor. How do the authors consider this factor in their study? Could the authors elaborate on this variable and the potential influence?
Author Response
Please see document attached.

Reviewer 3 Report
IJERPH 1033248
Thank you for the opportunity to review this manuscript. It is a well written manuscript and provides valuable information on the baseline of the Age.Well.de study
The aim is to carry out a multi-component intervention in a community older adult population without cognitive impairment in order to stop its appearance or minimize it.
I feel that the information on this subject is good but I think that it is low relevance as is presented since it is a descriptive study of the population according to groups. I believe that the authors should go a little deeper into the analyses made to provide more interesting data
Introduction,
I feel the authors have made a clear statement of the magnitude of the problem and its justification
Methods,
I think the authors should clarify the collection and classification of some variables:
How many GPs are involved in the assessments, were any inter- or intra-evaluation tests carried out?
What is considered low, intermidiate and high education?
How is physical activity evaluated? Only by times/week or by weekly time or type of exercise?
Drinking alcohol. How was this variable assessed?
Fruit and vegetable What instrument was used to calculate this variable?
How much time was needed for the initial assessment?
According to the bbliography, the authors could analyze which factors are more related to the cognitive deterioration and correlate them with the scales that evaluate them in order to know better the basal situation of the analyzed population and then avoid confusing factors in the final analysis after the intervention.
Results,
Tables, please review all units used for each variable (points, seconds, mmHg… etc)
Table 2 Second column is the same of third column of table 1
Discussion
It is simple, the lack of strong data to discuss is therefore observed in this section
Conclusions, Are correct
All the best in your submission!
Author Response
Please see document attached.

Round 2
Reviewer 1 Report
In the abstract and introduction please indicate the reason for presenting the methodology and baseline characteristics of the study population. Maybe to describe the population of at-risk subjects presumed able to consent to the intervention and potentially able to benefit? Please also comment on this in the conclusion.
It will be important to resolve the MOCA licensing issue prior to consideration for publication.
Reviewer 2 Report
I have no comments anymore. The authors have replied clearly and adequately to my suggestions.
Reviewer 3 Report
The authors have improved the manuscript, thanks for the effort
